# Transcriptome Analysis Revealed Hormone Pathways and bZIP Genes Responsive to Decapitation in Sunflower

**DOI:** 10.3390/genes13101737

**Published:** 2022-09-27

**Authors:** Lili Dong, Yu Wu, Jianbin Zhang, Xinyi Deng, Tian Wang

**Affiliations:** College of Horticulture, Anhui Agricultural University, Changjiang West Road, Shushan District, Hefei 230036, China

**Keywords:** sunflower, shoot branching, decapitation, hormone, bZIP, expression analysis

## Abstract

Decapitation is an essential agricultural practice and is a typical method for analyzing shoot branching. However, it is unclear exactly how decapitation controls branching. In this study, the decapitation of sunflower plants led to the development of lateral buds, accompanied by a decrease in indole-3-acetic acid (IAA) and abscisic acid (ABA) levels and an increase in cytokinin (CK) levels. Additionally, 82 members of the HabZIP family were discovered and categorized into 9 groups, using phylogenetic and conservative domain analysis. The intron/exon structure and motif compositions of HabZIP members were also investigated. Based on tissue-specific expression and expression analysis following decapitation derived from the transcriptome, several HabZIP members may be involved in controlling decapitation-induced bud outgrowth. Therefore, it is hypothesized that the dynamic variations in hormone levels, in conjunction with particular HabZIP genes, led to the development of axillary buds in sunflowers following decapitation.

## 1. Introduction

Sunflowers (*Helianthus annuus* L.) are one of the most important ornamental plants used for potted ornamental plants. During their cultivation, it is necessary to promote branching by means of decapitation. This substantially raises production costs. Consequently, developing new sunflower varieties with more branches is of enormous practical importance.

Apical dominance is one of the most common developmental phenomena in plants determining the architecture of the aboveground structure of plants [1]. The breakage of apical dominance by decapitation of the shoot tip leads to the outgrowth of dormant axillary buds to form branches. In the classical auxin application experiment, auxin placed on the decapitated stump of *Vicia faba* inhibited bud outgrowth, indicating that auxin plays a crucial role in the apical dominance process [2]. Auxin is synthesized in young growing leaves and is transported to plant stems through its specific polar auxin transport (PAT) stream but does not enter the buds. Therefore, auxin is considered to act indirectly on bud outgrowth [3]. Most findings indicate that auxin may restrict bud expansion by increasing strigolactone (SL) biosynthesis and suppressing cytokinin (CK) production [4].

Even in apical auxin, CK application to a bud can stimulate outgrowth [5]. However, the role of CK in regulating bud expansion remains unclear. Endogenous CKs can enter the axillary buds via acropetal transfer in the xylem sap. Consequently, CKs are the second messenger of auxin [6]. It has been postulated that CK controls early bud outgrowth by at least partially controlling the expression of the transcription factor *BRANCHED1* (BRC1), a bud outgrowth inhibitor [7]. CK is also believed to promote bud outgrowth by regulating auxin transport [8].

Studies have revealed a correlation between increased bud dormancy and higher amounts of abscisic acid (ABA) [9], while decapitation led to a drop in ABA levels in the buds [10]. Recent findings suggest that ABA is involved in the control of shoot-branching by means of light quality [11] and intensity [12] and that ABA acts downstream of *BRC1* [13]. Therefore, it is believed that ABA participates in the plant’s early reactions to decapitation and modulates bud dormancy in response to environmental conditions, such as shadow and low light intensity [3].

Numerous transcription factors, including the basic region/leucine zipper, have been identified as essential regulators of shoot-branching, according to previous studies (bZIP). The bZIP transcription factor family is one of the biggest, with members possessing a conserved bZIP domain composed of a primary region and a leucine zipper dimerization motif [14]. The highly conserved primary region of 16 amino acid residues with an invariant N-x7-R/K motif is crucial for nuclear localization and DNA binding. The leucine zipper is a dimerization motif consisting of a heptad repetition of leucines or other bulky hydrophobic amino acids, located nine units away from the C-terminus. Members of the bZIP family have been discovered in several plants, such as cucumber [15], cassava [16], tomato [17], chrysanthemum [18], etc. *CmbZIP1* was believed to adversely affect the branching of chrysanthemum shoots [18].

How decapitation causes the growth of lateral buds has not been fully explained. In this study, we detected changes in the auxin, cytokinin, and ABA contents, one and two days after decapitation was conducted in sunflowers. The bZIP family genes in sunflowers were identified, and their expression levels were analyzed. This work lays a theoretical foundation for further revealing the mechanism of shoot branching in sunflowers.

## 2. Materials and Methods

### 2.1. Plant Materials

The *H**. annuus* variety ‘Huoli’ was used in this study. The growth condition of sunflower seedlings was the same as in the description by Yao et al. [19].

### 2.2. Bud Length Measurements

For this study, 25-day-old sunflower seedlings were decapitated and the bud length was measured every two days with a vernier caliper or ruler, and photographs were also taken.

### 2.3. Hormone Quantification

For hormone quantification, 25-day-old sunflower seedlings were decapitated. The buds were harvested at 0 d, 1 d, and 2 d, respectively. The samples at each time point were biologically replicated three times and each biological replicate weighed at least 0.6 g. Then the samples were sent to Wuhan Metwell Biotechnology Co., Ltd. (Wuhan, China). The detection of endogenous IAA, CK, and ABA was conducted according to the procedure detailed by Bryksová et al. (2021) [20] and He et al. (2020) [21].

### 2.4. RNA Extraction, cDNA Library Construction, and SMRT Sequencing

RNA was extracted from the roots, stems, leaves, axillary buds, and flowers of 50-day-old sunflower seedlings, to investigate tissue-specific expression. For the experiment, 25-day-old seedlings were decapitated and the buds were taken for RNA extraction at 0, 1, and 2 days, respectively. Each tissue or treatment test had three biological repetitions.

RNA was isolated using TRNzol Universal Reagent (TIANGEN, Beijing, China) according to the manufacturer’s instructions. The integrity and amount of RNA were assessed using 1% agarose gel electrophoresis, while the purity and concentration were measured using a NanoDrop 2000 (Thermo Scientific, Waltham, MA, USA). Following the manufacturer’s instructions, libraries were prepared using a NEBNext^®^ Ultra^TM^ Directional RNA Library Prep Kit for Illumina (NEB, Boston, MA, USA). The mRNA was fragmented into 200-nt-long pieces by adding a fragmentation buffer. First-strand cDNA was synthesized using random hexamer-primed reverse transcription, followed by second-strand cDNA synthesis by DNA polymerase I and RNase H. After end-repair and adaptor ligation, the products were size-fractioned using agarose gel electrophoresis and then amplified using PCR. The Beijing Novogene Biological Information Technology Co., Ltd., (Beijing, China) used the Illumina TruSeq RNA Sample Preparation Kit to sequence the amplified fragments (Illumina, San Diego, CA, USA).

### 2.5. Sequencing Analysis and Differential Expression Analysis

The raw reads were trimmed, then the clean reads were obtained after removing the low-quality reads containing more than 50% bases with a Q-value of ≤10. The methods used for transcriptome assembly have been described previously [22]. The cleaned reads were mapped to the assembled sunflower transcriptome using Kallisto v.0.46.2. The number of mapped reads was computed and normalized for gene expression analysis using FPKM (the expected number of fragments per kilobase of transcript sequence, per million base pairs sequenced) [23]. The heat map was created with the ClustVis tool (https://biit.cs.utee/clustvis/, accessed on 3 March 2022) based on the average value of expression level of three biological repeats. EdgeR (version 3.16.5) was used to evaluate the differentially expressed genes (DEGs) to compensate for multiple testing when comparing two groups, while the false discovery rate (FDR) was utilized to calculate the *p*-value threshold. Genes with |log2 (FC)| ≥ 1 and FDR ≤ 0.05 were used as the significance threshold for gene expression differences.

### 2.6. Identification of HabZIP Genes

The HabZIP family databases of *Arabidopsis* were downloaded from the Arabidopsis Information Resource (TAIR9) (www.arabidopsis.org, accessed on 6 July 2022). HabZIP family member sequences were retrieved from the sunflower genome database (https://www.sunflowergenome.org/, accessed on 6 July 2022). Repetitive sequences were deleted, and Pfam (http://pfam.xfam.org/, accessed on 6 July 2022) was used to search the HabZIP family domain. We then removed the sequences that did not contain the bZIP domain and, finally, identified the HabZIP family member sequences. As detailed by Yao et al. [19], the prediction of the physical and chemical elements of HabZIP proteins was then performed.

### 2.7. Analysis of the Phylogenetic Tree, Motifs, and Gene Structure

The bZIP protein sequences of sunflowers and *Arabidopsis* were compared using MEGA 5.05 software. According to Yao et al. [19], a phylogenetic tree was constructed using a neighbor-joining method. The conserved motifs for *HabZIP**s* were analyzed with MEME Suite tools, while a gene structural analysis of *HabZIPs* was conducted using the Gene Structure Display Server (GSDS) software.

### 2.8. Statistical Analysis

All the data in this study were presented as the mean value ± SD. Tukey’s HSD tests were performed for statistical analysis.

## 3. Results

### 3.1. Effect of Decapitation on Bud Elongation in Sunflowers

Decapitation causes the outgrowth of axillary buds, which has been confirmed in pea plants [24], *Cremastra appendiculata* [25], and *Chimonanthus praecox* [26]. In sunflowers, two axillary buds sprouted simultaneously after removing the shoot tips. The length of the sunflower bud at the first node was measured for several days following decapitation. Figure 1 demonstrates that the average length of the buds rose from the second day following decapitation, up to 16 mm on the tenth day. However, the entire plant’s axillary buds did not expand appreciably.

### 3.2. Hormone Analysis

To find out whether endogenous hormones react to decapitation, the hormone composition of sunflower buds, including IAA, CKs, and ABA, was examined (Figure 2). Apparent decreases in the levels of IAA and ABA were observed at 1 d and 2 d after the decapitation treatment. IAA-Asp is a reversible storage form of IAA that can be reduced to free the IAA in plants [27]. Unlike IAA, the content of IAA-Asp showed an upward trend. Dihydrozeatin-7-glucoside (DHZ7G), dihydrozeatin-O-glucoside riboside (DHZROG), N6-isopentenyladenine (IP), and trans-zeatin-O-glucoside (tZOG) contents increased dramatically, regardless of whether it was one or two days after decapitation. The cis-zeatin riboside (cZR) and N6-isopentenyladenosine (IPR) contents dramatically increased one day after decapitation and then tended to stabilize. Dihydrozeatin ribonucleoside (DHZR), trans-zeatin (tZ), and trans-zeatin riboside (tZR) all showed a tendency to first increase and then decrease.

### 3.3. Transcriptome Analysis

To investigate the molecular processes that occurred in the sunflower buds following decapitation, nine RNA-seq libraries with three biological replicates, termed Decap 0 d, Decap 1 d, and Decap 2 d, were sequenced. In the gene expression analyses, the 0 d group served as the control. In total, 42.01 to 50.08 million raw readings were collected (Appendix A). Following removal of the adaptor sequences, poly N-containing sequences, and low-quality reads, over 41.47 million clean reads (6.22 G clean bases) were produced. The Q20 and Q30 percentages exceeded 96.7 and 91.74%, suggesting high sequencing quality. These sequences were then matched with the sunflower reference genome (https://sunflowergenome.org/, accessed on 2 February 2022). Over 82.67% of the clean reads were successfully matched to the reference genome; the unique matching rate was higher than 79.4%, with just 3.08 to 3.49% being linked to numerous places.

Comparing the transcriptomes of Decap 0 d, Decap 1 d, and Decap 2 d with Decap 0 d revealed the plant’s transcriptional responses to decapitation. A heatmap depicted global gene expression patterns when under several treatments (Figure 3A). A total of 8733 DEGs, including 4354 up- and 4379 down-regulated genes, were identified on day 1 after decapitation. In comparison, 11,958 DEGs, including 5680 up- and 6278 down-regulated genes, were identified on day 2 (Figure 3B,C), indicating that additional genes may be involved in the growth of axillary buds after the axillary bud germination signal is activated. Among the DEGs identified in this study, there are 2403 DEGs regulated explicitly on day one after decapitation, 5628 DEGs on day 2, and 6330 co-regulated genes that responded at 1 d and 2 d after decapitation (Figure 3D). We selected the bZIP transcription factor family for further analysis among these genes, showing significant changes.

### 3.4. Genome-Wide Identification of bZIP Members in Sunflowers

From the sunflower, 82 potential bZIP members have been identified and the subsequent identification of conserved domains verified that all detected bZIPs include the conserved bZIP domain. The 82 predicted full-length HabZIP proteins ranged from 105 (HabZIP22) to 584 (HabZIP30) amino acid (aa) residues, from 11,941.69 (HabZIP60) to 63,658.61 (HabZIP30) Da in relative molecular mass, and from 5.1 (HabZIP68) to 11.3 (HabZIP22) in isoelectric points (Appendix A).

### 3.5. Classification of HabZIPs by Phylogeny and the Identification of Domain Conservation

To investigate the evolutionary connections and categorization of the bZIP family, a phylogenetic tree comprising 82 HabZIPs and 78 *Arabidopsis* members was created. As shown in Figure 4, we divided the sunflower bZIP proteins into nine groups according to the grouping of *Arabidopsis*. The size of the nine groups varies; the two largest groups have 18 (S group) and 17 (A group) members, while group H has only two.

### 3.6. HabZIPs Gene Structure and Conserved Motifs

To gain a deeper understanding of the structural characteristics of the HabZIP genes, we investigated each member’s intron/exon structure and motif makeup (Figure 5). Some individuals of the same group share similar gene structures, as anticipated. For instance, four out of six members of the D group include two introns. Each member of group H possesses three introns. In contrast to other subfamilies, the gene organization of some subfamilies varies significantly.

Ten conserved motifs were identified, motif 1 being the bZIP domain and motifs 2 and 4 being DOG1. As predicted, all members of HabZIP share motif 1. Motif 2 is most prevalent in groups D, E, and I. Motifs 5 and 6 are predominant among group A members. Motif 4 is present only in groups S and C. Motifs in particular groups may be associated with particular biological activities.

### 3.7. Analysis of HabZIP Gene Expression Patterns in Five Tissues

Based on the transcriptome data from roots, stems, leaves, buds, and flowers, the expression profiles of HabZIPs were examined to obtain insight into the bZIP transcription factor expression patterns in sunflowers (Figure 6). In total, 12 genes were not detected; the detected genes showed different tissue specificity. There were 15, 16, 13, and 4 genes found to be highly expressed in the flower, root, stem, and leaf, respectively. However, we are most concerned about the genes with the highest expression levels in the leaf axils, including *HabZIP20*, *HabZIP47*, *HabZIP50*, *HabZIP14*, *HabZIP40*, *HabZIP67*, *HabZIP39*, *HabZIP28*, *HabZIP64*, *HabZIP29*, *HabZIP30*, *HabZIP69*, *HabZIP59*, *HabZIP74*, *HabZIP62*, *HabZIP55*, *HabZIP78*, *HabZIP73*, *HabZIP70*, *HabZIP42*, *HabZIP43*, and *HabZIP53*.

### 3.8. Expression Profiles of HabZIP Genes Responding to Decapitation

To explore the potential functions of HabZIP genes that are involved in shoot branching, we further analyzed the expression characteristics of the HabZIP genes that expressed highest in buds after decapitation. As we can see from Figure 7, seven genes were significantly up-regulated after decapitation, among which the expression levels of *HabZIP47*, *HabZIP40*, *HabZIP78*, *HabZIP73*, *HabZIP53*, and *HabZIP62* were continuously up-regulated, while *HabZIP29* first increased and then decreased. The expression level of 13 genes decreased significantly. Among them, the expression levels of *HabZIP64*, *HabZIP70*, and *HabZIP42* continued to decline, while *HabZIP39*, *HabZIP28*, *HabZIP30*, *HabZIP74*, *HabZIP55*, *HabZIP50*, *HabZIP43*, *HabZIP20*, *HabZIP14*, and *HabZIP67* first decreased and then increased. In addition, the expression levels of *HabZIP69* and *HabZIP59* did not change significantly one day after decapitation but significantly increased on the day after that.

## 4. Discussion

In plants, decapitation triggers bud outgrowth, while auxin application to the decapitated stump is considered to result in the maintenance of apical dominance. However, in some species, auxin application cannot fully restore apical dominance [28]. These results suggest that shoot-tip removal is not equivalent to removing the auxin source, and decapitation may activate other downstream signals or pathways. Therefore, exploring these downstream pathways or signaling molecules has become an exciting direction.

We detected the length of axillary buds at different times after decapitation. It was found that the growth trend and final lengths of the two opposite axillary buds were approximately the same, and the apparent growth of the sunflower’s axillary buds could be seen two days after decapitation. The level of IAA decreased significantly one day after decapitation but increased slightly two days after decapitation. This result is similar to the study in *C**. appendiculata* [25]. The declining IAA content was accompanied by an accumulation of IAA-Asp, confirming a low ratio of IAA to IAA-Asp in the axillary buds after decapitation. The germination of axillary buds after decapitation was closely related to the decreasing synthesis of ABA, which is consistent with the study conducted using woody bamboos [29]. In addition, ABA showed a continuous downward trend after decapitation, implying that perhaps ABA not only plays a role in the germination of axillary buds but also in the subsequent bud growth.

One day after decapitation, the contents of DHZ7G, DHZROG, cZR, tZOG, DHZR, tZ, tZR, IPR, and IP all dramatically rose. Similarly, a significant increase in tZR, IP, and IPR was observed in decapitated rice [30], while the remarkably increased accumulation of IP, tZ, IPR, tZR, and DHZR was detected in the tomato *dwf* mutant [31]. The above results show that tZR, IP, IPR, tZ, and DHZR are closely related to bud germination after decapitation. Two days after decapitation, the IP and IPR continued to increase or remain stable, indicating that they may continue to participate in the subsequent growth of axillary buds after germination. The tZOG content dramatically rises, regardless of one or two days after decapitation, while the overexpression of *ZOG1* resulted in the increase of tZOG and the decreasing of apical dominance in *Phaseolus lunatus* [32], indicating that tZOG also plays a vital role in axillary bud development. In addition, the contents of cZR, DHZ7G, and DHZROG showed an apparent response to decapitation, implying that these three substances may also be involved in shoot branching.

We found 82 HabZIP family members during this study. Previous research has found 75 bZIPs in *Arabidopsis*, 89 in rice, 64 in cucumber, 131 in soybean [33], and 96 in *Brachypodium distachyon* [34]. These results revealed that bZIP numbers in sunflowers had increased relative to Arabidopsis, cucumber, and grapevine, whereas they had decreased relative to maize, soybean, and *B. distachyon*. In contrast to the earlier evolutionary categorization of bZIPs [16], the evolutionary study revealed that HabZIPs might be categorized into nine subfamilies. This categorization was further validated by our analyses of gene structure and conserved motifs. Analysis of the gene structure revealed that the number of introns in HabZIPs ranged from 0 to 12 (see Figure 5). The number of introns in grapevine and *B. distachyon* ranged from 0 to 10 and from 0 to 13, respectively [16]. This indicated that the structural diversity of bZIP genes across animals was comparable. The analysis of conserved motifs revealed that nearly all the HabZIPs had a conventional bZIP domain. In addition, each subfamily has specific shared motifs, while some subfamilies also possessed unique motifs (Figure 5). These characteristics were also seen in the conserved motifs of grapevine bZIPs [16]. Most HabZIP genes within the same subfamilies generally had similar gene structures and conserved motifs, supporting their tight evolutionary connection and subfamily categorization.

Each HabZIP gene’s profile of expression in five distinct tissues was elucidated. Some HabZIP genes were not discovered in any of these five organs, presumably due to expression patterns that cannot be analyzed using our databases. Most HabZIP genes are strongly expressed in the leaf axils and flowers, suggesting that these genes play a crucial role in axillary bud and flower formation. We further analyzed the expression of genes with higher expression levels in axillary buds after decapitation. Most of these genes were significantly upregulated or downregulated after decapitation. These results suggest that these genes may positively or negatively regulate the shoot-branching of sunflowers. The expression levels of some genes were continuously down-regulated or significantly up-regulated on the second day after decapitation, indicating that these genes may also play an essential role in the subsequent growth of axillary buds after activation.

Based on the above studies, we found that the contents of IAA, CK, and ABA changed significantly after decapitation, and the expression levels of the HabZIP genes were significantly changed, which indicated that the decapitation caused axillary bud germination, at least partially, by regulating the changes in the hormone pathway and the expression levels of bZIP genes. This study provides an experimental and theoretical basis for revealing the branching mechanism of sunflower plants.

## 5. Conclusions

Decapitation can cause the germination of axillary buds, but the specific mechanism is not completely clear. In this study, we found that decapitation caused a decrease in auxin synthesis and an increase in cytokinin synthesis. In addition, a total of 82 HabZIP genes were identified in the sunflower genome. These genes were divided into 9 subfamilies, and the basic characteristics, gene structures, conserved motifs of these genes were analyzed in detail. Gene expression analysis based on RNA-seq indicated that some HabZIP members played important roles in shoot branching. These results provide a theoretical basis for further revealing the molecular mechanism of axillary bud germination caused by decapitation and exploring the function of HaZIP members.

## Figures and Tables

**Figure 1 genes-13-01737-f001:**
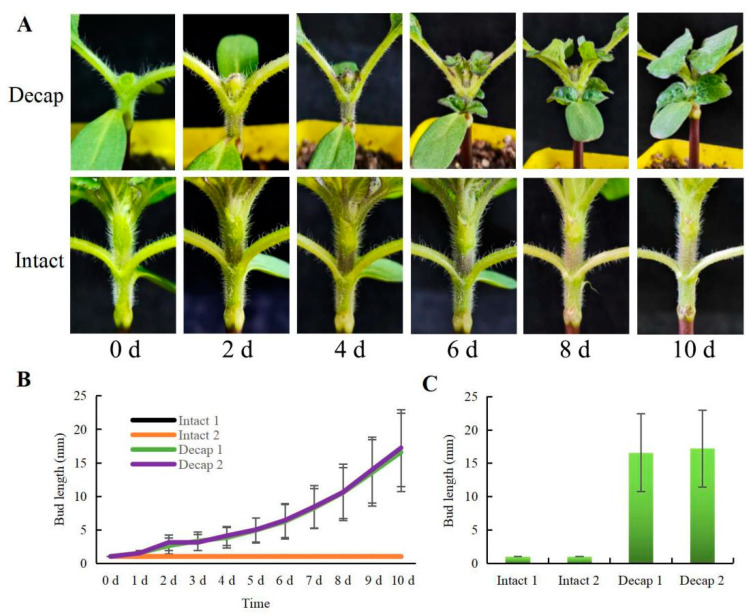
Effects of decapitation on the bud growth of sunflowers. (**A**) Sunflower buds after decapitation and the control group (Intact). (**B**) The bud length of sunflowers on different days after decapitation (*n* = 12). (**C**) The bud length of the sunflowers on the 10th day (*n* = 12). Values are means ± SE for (**B**,**C**).

**Figure 2 genes-13-01737-f002:**
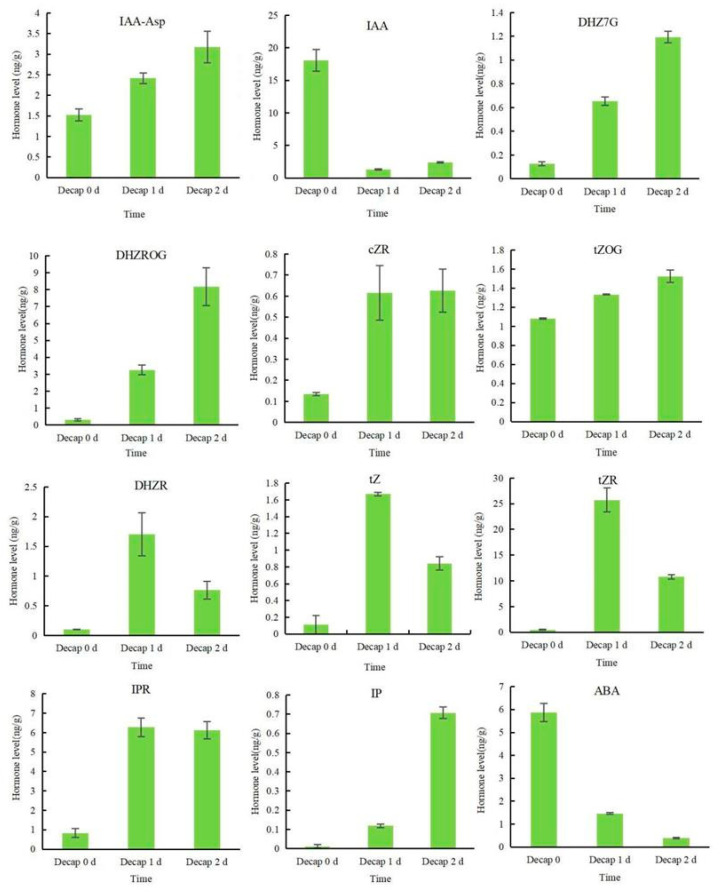
Effects of decapitation on the hormone levels of sunflower flower buds. Data indicate the means ± SE (*n* = 20 plants).

**Figure 3 genes-13-01737-f003:**
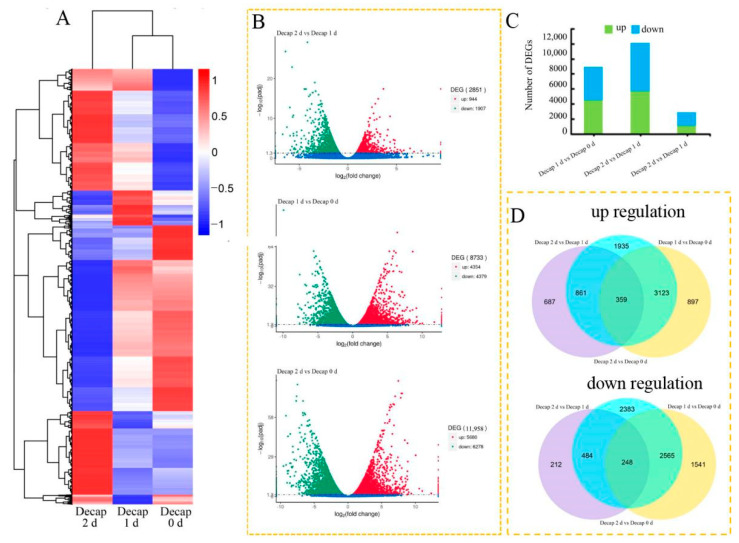
Transcriptional variations in sunflower buds after decapitation. (**A**) A heatmap shows the expression profiles of the DEGs under different treatments. (**B**) Analysis of the significance of the DEGs in various comparisons using volcano plots. (**C**) The number of genes that were up-and down-regulated in the various comparisons. (**D**) Venn diagrams illustrating the proportions of up-and down-regulated genes in three contrasts.

**Figure 4 genes-13-01737-f004:**
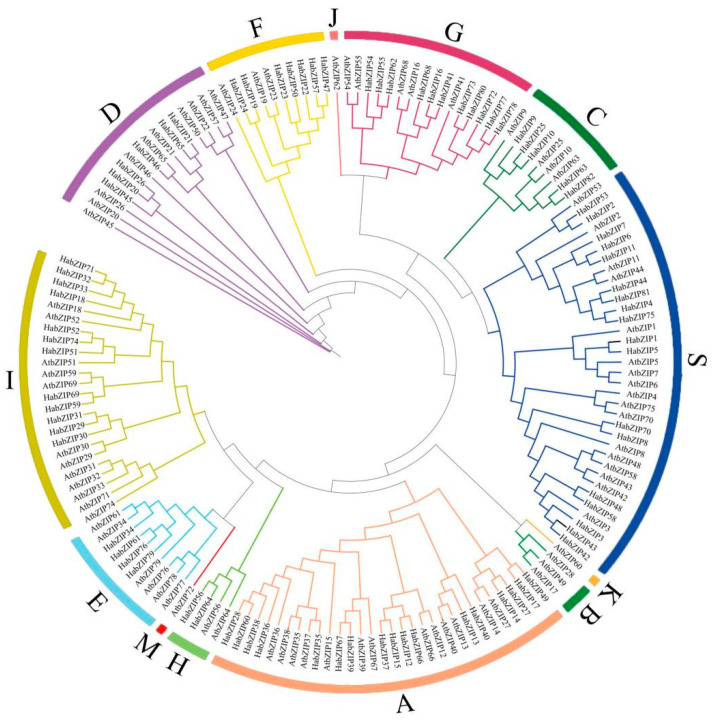
Phylogenetic tree of the bZIPs from sunflowers and *Arabidopsis*. The capital letters represent different groups.

**Figure 5 genes-13-01737-f005:**
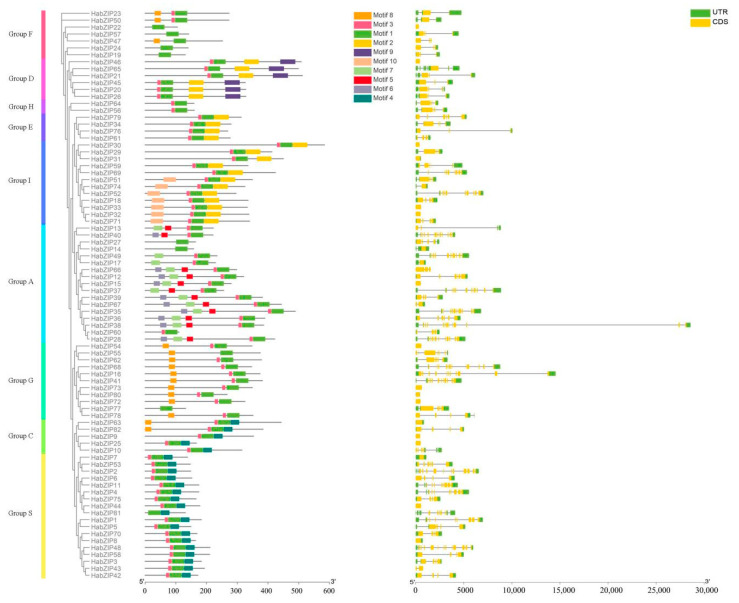
Sequence analysis and gene structure of HabZIPs.

**Figure 6 genes-13-01737-f006:**
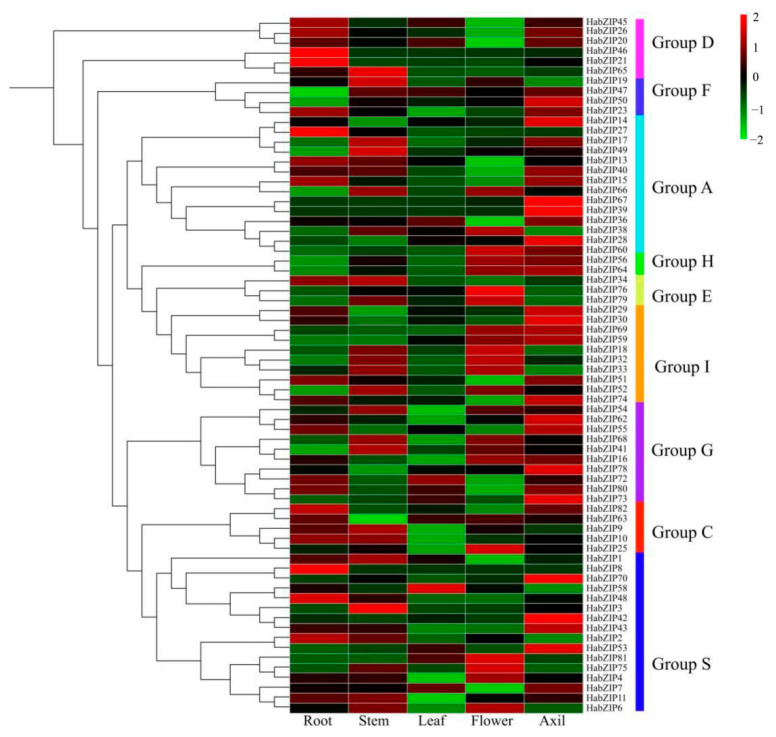
HabZIP gene expression analysis in five sunflower tissues. The transcript abundance of HabZIP genes is represented by the color scale (−2 to 2) and the latter’s heat map is the same as this.

**Figure 7 genes-13-01737-f007:**
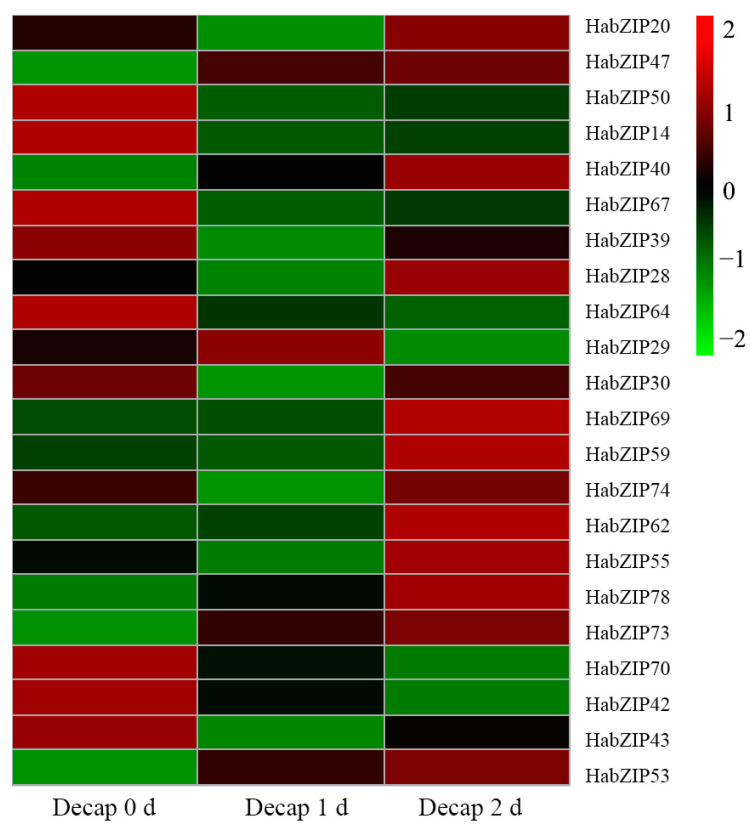
Response of the HabZIP genes to decapitation.

## Data Availability

Not applicable.

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
