# Peer review of "Transcriptome Analysis Revealed Hormone Pathways and bZIP Genes Responsive to Decapitation in Sunflower"

_genes, 2022, doi:10.3390/genes13101737_

Round 1

Reviewer 1 Report

Dear authors,   In the manuscript Transcriptome analysis revealed hormone pathways and bZIP genes responsive to decapitation in Sunflower” submitted to Genes, the authors studied hormone pathways and bZIP genes responsive to decapitation in Sunflower.    Overall, the presented rationale, methodology and conclusions scientifically well established. However, I have some major concerns about the methods and results presented by the authors. In particular the methods section is very rudimentary, which makes the presented results and discussion extremely difficult to follow, interpret and justify. Moreover, the authors not mention the use of any replicates, which makes the results less robust.   Below, there are comments which need to be explained in more detail:   L72: Did the authors use biological or technical replicates? There is no information whether replicates were used at all. L90: How were the reads cleaned? L90: the assembled sunflower transcriptome: Did the authors assemble the transcriptome themselves? If so, how was this done? L91: Tophat is already quite old. There are newer mappers (like kallisto, salmon) available  L96: What version of edgeR was used? L104-106: What was the rational behind this? Please check the grammar of the sentence and rephrase it L111: What version of MEGA was used? L156: There shouldn’t be a space between the sample 0,1,2, and d.  L159: There’s no such table in the supplement  L164: There’s is no method describing the read mapping against the sunflower genome, only against the transcriptome. What version of the assembly was used? What annotation version? L165: What is higher? 100%? L169: What is the total number of genes? What is the percentage of DEGs vs total gene number? L176: How were the bZip identified?  L234: What does the legend show? What is 2 to -2? L247: What does the legend show? What is 2 to -2? L281: Did the authors identify the 82 HapZIP members in sunflower themselves or have these been annotated before in the published sunflower genome? (https://doi.org/10.1038/nature22380) L283: Please use numbers

Author Response

L72: Did the authors use biological or technical replicates? There is no information whether replicates were used at all. 

We have added it.

L90: How were the reads cleaned? 

We have added it.

L90: the assembled sunflower transcriptome: Did the authors assemble the transcriptome themselves? If so, how was this done? 

We have added the reference.

L91: Tophat is already quite old. There are newer mappers (like kallisto, salmon) available  

We have revised it.

L96: What version of edgeR was used? 

We have added it.

L104-106: What was the rational behind this? Please check the grammar of the sentence and rephrase it 

We have revised it.

L111: What version of MEGA was used? 

We have revised it.

L156: There shouldn’t be a space between the sample 0,1,2, and d.  

OK

L159: There’s no such table in the supplement  

We have uploaded the Supplementary Table S1.

L164: There’s is no method describing the read mapping against the sunflower genome, only against the transcriptome. What version of the assembly was used? What annotation version? 

We have added the reference.

L165: What is higher? 100%? 

We have revised it.

L169: What is the total number of genes? What is the percentage of DEGs vs total gene number? 

L176: How were the bZip identified?  

L234: What does the legend show? What is 2 to -2? 

We have added it.

L247: What does the legend show? What is 2 to -2? 

We have added it.

L281: Did the authors identify the 82 HapZIP members in sunflower themselves or have these been annotated before in the published sunflower genome? (https://doi.org/10.1038/nature22380) 

We identified the 82 HapZIP members ourselves.

L283: Please use numbers

We have added it.

Reviewer 2 Report

The manuscript “Transcriptome analysis revealed hormone pathways and bZIP genes responsive to decapitation in Sunflower” identifies some of the HabZIP members which respond to decapitation-induced shoot branching using transcriptome analysis. This work is based on the hypothesis that the combinatorial effect of hormonal variations and alteration of HabZIP genes regulate the decapitation-induced development of axillary buds in sunflowers. The manuscript gives preliminary information on the alteration in the expression profiling of the HabZIP genes following the decapitation process along with the endogenous hormonal changes. The authors have not indicated any correlation between the two factors. However, the manuscript lacks the experimental analysis to justify the involvement and role of the bZIP family of genes in the process.

Please follow the comments:

1.      Materials and methods: The method section lacks statistical analysis. Kindly indicate the number of variables used for the measurement of parameters in each case.

2.      Indicate the biological and technical samples used for the transcriptome analysis of the various tissues and buds following decapitation.

3.      Indicate how the nomenclature of the HabZIP family was performed? The authors indicate the Arabidopsis bZIP family included 75 members and 82 in case of sunflower. Kindly mention the members of the bZIP family not found in Arabidopsis.

4.      The manuscript solely relies on the transcriptome analysis data, however, there is no mention of any experimental validation of the expressional pattern of the DEGs indicated to play a role in the decapitation process.

5.      The authors could have also indicated the effect of the external hormonal application on the buds following decapitation to justify the hormonal influence on decapitation in the case of sunflower. Indicate why the authors have not correlated the endogenous and external effects of the identified hormonal variation (auxin/cytokinin/ABA) in the decapitation study.

6.      How was the heat map generated for the DEGs? Kindly indicate if the expression profiling was indicated as mean values of the transcriptome data for each tissue.

Author Response

1 Materials and methods: The method section lacks statistical analysis. Kindly indicate the number of variables used for the measurement of parameters in each case.

We have added it.

2 Indicate the biological and technical samples used for the transcriptome analysis of the various tissues and buds following decapitation.

We have revised it.

3 Indicate how the nomenclature of the HabZIP family was performed? The authors indicate the Arabidopsis bZIP family included 75 members and 82 in case of sunflower. Kindly mention the members of the bZIP family not found in Arabidopsis.

It should be 78 members in Arabidopsis, and we have made changes. The number of 78 and 82 is not much different. After a careful check, there are no rules. Some subfamilies have more members in petunia, while some subfamilies have more members in Arabidopsis. Therefore, we did not added any more explanations here.

4 The manuscript solely relies on the transcriptome analysis data, however, there is no mention of any experimental validation of the expressional pattern of the DEGs indicated to play a role in the decapitation process.

In this study, we just preliminarily analyze the members who may participate in decapitation response through transcriptome data, and then we will continue to study the functions of these screened genes with another article.

5 The authors could have also indicated the effect of the external hormonal application on the buds following decapitation to justify the hormonal influence on decapitation in the case of sunflower. Indicate why the authors have not correlated the endogenous and external effects of the identified hormonal variation (auxin/cytokinin/ABA) in the decapitation study.

There have been many studies on the effects of auxin and cytokinin on axillary bud outgrowth, which are relatively basic experiments. We think it is unnecessary to apply exogenous hormones. ABA has no obvious promoting effect on axillary buds, and endogenous ABA may play a key role.

6 How was the heat map generated for the DEGs? Kindly indicate if the expression profiling was indicated as mean values of the transcriptome data for each tissue.

Yes, we did the heat map by taking the average of three duplicates

Round 2

Reviewer 2 Report

The authors have revised the manuscript as per the suggestions.

The authors report the hormonal involvement and genes belonging to the bZIP family related to decapitation followed by axillary bud development in sunflowers. The transcriptome data indicate the important bZIP genes which can be further studied to understand the mechanism involving decapitation leading to axillary bud development in sunflower and related families.

Apart from the above positive points, the manuscript lacks experimental evidence related to the transcriptome datasets to further justify their outcomes and is beyond the scope of the manuscript in the current form. The correlation between two different parameters i.e. the study of hormones and the genes belonging to the bZIP family is not shown in the current manuscript. The transcriptome analysis could have been focused on studying and characterizing the hormonal alterations in depth which is not shown in the manuscript.

However, the transcriptome analysis performed provides preliminary but useful information as a basis for future validation of the gene networks involved in the process. The same observation goes for the hormonal analysis performed.